# Learning to Understand: Incorporating Local Contexts with Global Attention for Sentiment Classification

**Zhigang Yuan & Yuting Hu**
Department of Electronic Engineering
Tsinghua University
Beijing, China
`{yuanzg14, hu-yt12}@mails.tsinghua.edu.cn`

**Yongfeng Huang**
Tsinghua National Laboratory for Information Science and Technology
Department of Electronic Engineering
Tsinghua University
Beijing, China
`yfhuang@tsinghua.edu.cn`

## Abstract

Recurrent neural networks have shown their ability to construct sentence or paragraph representations. Variants such as LSTM overcome the problem of vanishing gradients to some degree, thus being able to model long-time dependency. Still, these recurrent based models lack the ability of capturing complex semantic compositions. To address this problem, we propose a model which can incorporate local contexts with the guide of global context attention. Both the local and global contexts are obtained through LSTM networks. The working procedure of this model is just like how we human beings read a text and then answer a related question. Empirical studies show that the proposed model can achieve state of the art on some benchmark datasets. Attention visualization also verifies our intuition. Meanwhile, this model does not need pretrained embeddings to get good results.

## 1 Introduction

Sentiment classification, a key problem in sentiment analysis, has drawn lots of interest since 2000s (Pang & Lee, 2008). Traditional machine learning based methods often use one-hot features to represent a text. In these methods, each word (or n-gram) is treated as an independent token, and a text is represented as the words within. Though convenient intuitively, these methods cannot model the semantic dependencies between words, and often lead to the curse of dimensionality problem.

Neural models have shown their great performance in learning representation. Bengio et al. (2003) proposed the feed forward neural network language model. Ever since, many neural network based models have been widely used in many natural language processing (NLP) tasks, such as named entity recognition (Collobert & Weston, 2008), machine translation (Cho et al., 2014) and sentiment analysis (dos Santos & Gatti, 2014). Compared to one-hot representation, words are represented as distributed, dense vectors (also called word embeddings) in these neural models. Mikolov et al. (2013b) showed that, through proper training, these vectors can model syntactic and semantic relationships between words. More importantly, these vectors can be calculated naturally, which leads to another problem — semantic composition.

For many NLP tasks, we often face the problem of representing a sentence or paragragh using word embeddings of its containing words. Simply methods such as concatenation or weighted sum cannot give us satisfactory results. Many neural models, such as recursive neural network (Socher et al., 2011), recurrent neural network (Mikolov et al., 2010) and convolutional neural networks (Collobert & Weston, 2008) have been proposed to solve this problem.

Since a sentence or paragraph is naturally a sequence of word tokens, recurrent neural networks (RNN) can be used to composite word embeddings into a sentence or paragraph embedding. Recurrent networks utilize special units which are connected to themselves to maintain a hidden state. The internal hidden state can be seen as a representation of the contexts. However, basic recurrent network suffers from the vanishing or exploding gradients problem (Hochreiter et al., 2001), due to its direct connection between states in adjacent timesteps. Long short term memory (LSTM) network alleviates this problem by introducing a cell that has the ability to read, write or reset its internal state (Graves, 2012). Even so, LSTM has the bias of prefering recent inputs, making it difficult to capture the important information within a long sequence.

Recently, many studies have been done to help RNNs improve their semantic composition performance. A common way is to introduce another kind of network, such as CNN into RNN (Lai et al., 2015). In these combined models, different parts share complementary advantages. Lately, the attention mechanism (Bahdanau et al., 2014) has drawn a lot of attention. In an attention model, we utilize states in all timesteps, rather than simply force the network to represent the context into one fixed-length vector. More importantly, by visualizing the attention weights, we can get a deeper insight of how the network works (Bahdanau et al., 2014; Rocktäschel et al., 2015).

Inspired by the attention thought, we propose a framework which is similar to how we human beings read a text and then answer an related question. This framework consists of two parts: the first part is a bidirectional LSTM (Bi-LSTM) to learn a rough representation of the whole text, the second part is another Bi-LSTM with attention, which can learn local contexts and incorporate them to get a refined context representation.

## 2 RELATED WORK

In this section, we introduce the related work concerning sentiment analysis, neural semantic composition and attention mechanism.

### 2.1 SENTIMENT ANALYSIS

Sentiment analysis, also called opinion mining or subjectivity analysis, is the field of study that analyzes people's sentiments or emotions towards various entities (Liu, 2012). Due to its great importance, sentiment analysis has a wide range of applications, such as finding opinions of certain products, predicting the stock market or political elections.

Generally, existing methods for sentiment classification fall into two categories: machine-learning-based methods and semantic-based methods. Semantic methods relies more on lexical and linguistic resources. Pang et al. (2002) first tried using machine learning methods to classifiy texts based on their sentiments. Ever since, a lot more classification models and features have been tested. For machine learning procedures, feature design is always the key step for a good result. While in most cases, a well-designed feature needs much human effort and domain knowledge, which severely limits its applications.

Deep learning models have drawn lots of attention due to their automatic representation learning. Deep learning is about learning multiple levels of representations, obtained by combining non-linear modules (LeCun et al., 2015). Various models have been tested for sentiment analysis task, such as recursive autoencoder (Socher et al., 2011), paragraph vector (Le & Mikolov, 2014), convolutional network (Kim, 2014), etc.

### 2.2 NEURAL LANGUAGE MODELS AND SEMANTIC COMPOSITION

Bengio et al. (2003) proposed a feed-forward neural network language model, which used distributed representations for words. With the rapid progress of neural language models for NLP tasks, the distributed representations of words have become more and more important.

In most neural language models, the words (or even characters) are represented as dense, distributed vectors. These embeddings are usually used as the inputs of a network. The subsequent layers then conduct semantic composition operations over these word/character embeddings to obtain higher level semantic represetations. This kind of representations based on embeddings of different seman-

tic levels (character, word, phrase, sentence, paragraph, etc.) have shown their great performance on many NLP tasks.

After Bengio's pioneering work, Collobert & Weston (2008) replaced the probabilistic output with a scoring scheme, and provided a unified architecture for many NLP tasks. Mnih & Hinton (2007) used a log-bilinear model to get the hidden representations. They future provided a hierarchical version (Mnih & Hinton, 2009), which uses a hierarchical softmax at the output layer to reduce calculation amount. Huang et al. (2012) added the global context information into C&W model, which can improve word embeddings as well as solve the polysemy problem. Their idea of using global context inspired our work, while in our work we use the global context as attention for the following training. Socher et al. (2011; 2012; 2013) proposed a family of recursive autoencoder models, which construct higher level represetations using a tree structure. Mikolov et al. (2013a) provided their word2vec (CBOW and Skip-gram) model. The word2vec model is actually a simple but effective log-linear model, imbibing advantages of previous neural models. Word embeddings pretrained using word2vec on a large corpus can be used for many other NLP tasks, often used as initializations or auxiliary features. Pennington et al. (2014) performed the training on aggregated global word-word co-occurrence statistics from a corpus, and their resulting global vectors (GloVe) showcase interesting linear substructures of the word vector space. Le & Mikolov (2014) extended the idea of word2vec model from word to paragragh, and the paragraph vectors they got can improve the performance on various NLP tasks.

Although seemingly different, these models all conduct semantic compositions directly or indirectly at intermediate layers. Currently, various neural models have been tried for semantic composition, such as multi-layer perceptron, recusive network, convolutional network and recurrent network. When representing higher level language units, convolutinal netwoks are better at capturing local information, while recurrent networks excel at storing history information. The network structure has also been explored, such as sequencial or tree structure (Li et al., 2015).

Of all these models, recurrent networks are highly valued due to its human alike reading style. More importantly, recurrent networks are capable of modeling sequential inputs or outputs, which is essential for tasks such as machine translation (Cho et al., 2014). When applying recurrent networks for NLP tasks, it is typical to encode the semantic information of all words within a sentence into one fixed-length vector. The resulting representation is then used for generating related output words or simply used for classification.

## 2.3 ATTENTION MECHANISM

Attention mechanism is first used for machine translation (Bahdanau et al., 2014) in a encoder-decoder framework. The key idea of attention is to allow the network revisit all parts of a source sentence for an output decision, instead of trying to encode all information of a source sentence into a fixed-length vector. By this mechanism, we can get a deeper insight of how the network works. In the work of Bahdanau et al. (2014), they found the alignment between words by visualizing the attention weights.

Successive studies on attention mechanism include image caption (Xu et al., 2015), semantic entailment (Rocktäschel et al., 2015), aspect level sentiment analysis (Tang et al., 2016), etc. The most related work to ours is an attention model for sequence classification (Shen & Lee, 2016), in which they used LSTM output to weight word embeddings. While in our work, we use the global context to selectively choose the most important local contexts, which provides mutual promotions for both global and local context representations.

## 3 LEARNING TO UNDERSTAND: THE MODEL

It's widely accepted that RNN alone, even with LSTM cells, cannot store long-time information. But, we may ask, do we really have to get a refined document or text embedding? Taking the reading comprehension task as an example, after a quick look at the text given, what we human beings got is also a rough memory of this text. Even so, this did not affect us to make a right decision. What we typically do next is to look at the problem, and rescan the text with a rough memory of the whole text in mind. With this global context information, we can easily locate the important parts for our

problem. In the end, even if we have not understood the whole text thoroughly, we can make right decisions for the problem in hand.

This thought inspired us to propose this attention model, which incorporates local contexts with a rough global context attention.

## 3.1 Basic Model: Bi-LSTM

Long short term memory network can be seen as a variant of the basic recurrent network, mainly to address the long-time dependency probem. It has special units that can pass their states to the next timestep. LSTM network replaces the hidden units with more flexible cells. The cell itself is a small network, which can decide its own reading, writing or resetting behavior.

The LSTM network we use is similar to Graves (2012). We don't consider peephole connections. Given an input sequence $\mathbf{s} = (x_1, x_2, \cdots, x_n), (x_i \in \mathcal{R}^m)$, the update procedure from timestep $t - 1$ to $t$ can be described as follows:

$$
\begin{align}
i_t &= \sigma(W^{ix}x_t + U^{ih}h_{t-1} + b^i) \tag{1}\\
f_t &= \sigma(W^{fx}x_t + U^{fh}h_{t-1} + b^f) \tag{2}\\
o_t &= \sigma(W^{ox}x_t + U^{oh}h_{t-1} + b^o) \tag{3}\\
\tilde{C}_t &= \phi(W^{cx}x_t + U^{ch}h_{t-1} + b^c) \tag{4}\\
C_t &= f_t \odot C_{t-1} + i_t \odot \tilde{C}_t \tag{5}\\
h_t &= o_t \odot \phi(c_t) \tag{6}
\end{align}
$$

where $\odot$ denotes the element-wise multiplication, $i_t, f_t, o_t \in \mathcal{R}^h$ stands for the input gate, forget gate, output gate respectively. The LSTM cell maintains an internal cell state. In one timestep, the input gate determines how much the outside information can come in, the output gate determines how much the internal information can go out, and the forget gate determines how much the internal information should be forgotten when passed to next timestep.

For a given sequence $\mathbf{s} = (x_1, x_2, \cdots, x_n)$, if we feed the token from left to right one by one into the LSTM network, the final internal state $\overrightarrow{h}$ will contain more information of the latter tokens due to the model bias. We feed the sequences backwards into the LSTM again, thus getting a backward-propagating internal state $\overleftarrow{h}$. The final representation of the global context is calculated by

$$
h = \overrightarrow{h} \oplus \overleftarrow{h} \tag{7}
$$

where $\oplus$ denotes concatenation. This is a typical bidirectional LSTM (Bi-LSTM) network. Again, we point out that this gloal context representation may be just a rough one. We use it as attention rather than directly treat it as a refined representation for the task afterwards.

## 3.2 Two-Scan Approach with Attention

In this part, we introduce our **T**wo-**S**can Approach with **Att**ention (TS-ATT) model.

The TS-ATT model contains two parts: a Bi-LSTM network first scans the whole text to get a global context represetation, another Bi-LSTM with attention scans the text again to get local contexts around each token, and composite them using the global context attention. The model are shown in Figure (1).

**First Scan**: we first use a Bi-LSTM network to gain a global context representation. As described in Sec. (3.1), the final global context vector $h$ is a concatention of the final output of left-LSTM $\overleftarrow{h}$ and right-LSTM $\overrightarrow{h}$. This global context serves as attention for the second scan. To distinguish the final context representation in the second scan, we denote $h$ as $h^{\texttt{scan1}}$, which is

$$
h^{\texttt{scan1}} = \overleftarrow{h} \oplus \overrightarrow{h} \tag{8}
$$

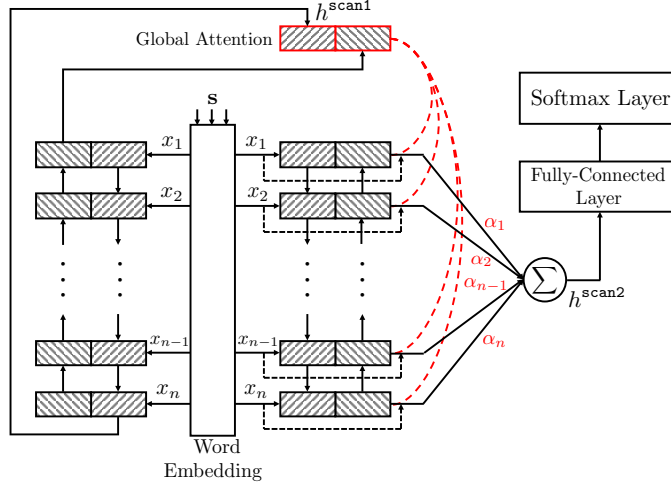

Figure 1: Two-Scan Approach Attention Model (TS-ATT)

**Second Scan**: During the second scan of the text, we use another Bi-LSTM networks to learn local context for each word. Since already with the rough global context in mind, when looking at one word, we actually see its context meaning rather than the word itself. Thus, we use a word's local context vector instead of its word embedding here. Inspired by the work of Lai et al. (2015), for word $w_i$, we concatenate its left context $c_i^{(l)}$, right context $c_i^{(r)}$ and embedding $e_i$ to construct its final local context represetation $c_i$, which is:

$$c_i^{(l)} = f\left(W^{(lc)}c_{i+1}^{(l)} + W^{(le)}e_i\right) \tag{9}$$

$$c_i^{(r)} = f\left(W^{(rc)}c_{i-1}^{(r)} + W^{(re)}e_i\right) \tag{10}$$

$$c_i = c_i^{(l)} \oplus e_i \oplus c_i^{(r)} \tag{11}$$

**Attention**: As we may expect, during the second scan, we don't have to encode every word into the text representation. Just paying attention to the parts which are important for our problem in hand would be enough. Thus, we use the global context vector $h^{\mathtt{scan1}}$ as the attention of all the local contexts. Attention weights are acquired by one-layer feed-forward network, which is:

$$y_i^{\mathtt{att}} = f\left(W^{\mathtt{att}}(h^{\mathtt{scan1}} \oplus c_i) + b^{\mathtt{att}}\right) \tag{12}$$

where $W^{\mathtt{att}}$ and $b^{\mathtt{att}}$ are attention parameters. All the attention weights are then fed into a softmax layer to obtain probabilistic attention weights. The final representation used for classification (denoted as $h^{\mathtt{scan2}}$) is the weighted sum of all the local context vectors, which is:

$$\alpha_i = \frac{\exp(y_i^{(a)})}{\sum_{i=1}^n \exp(y_i^{(a)})} \tag{13}$$

$$h^{\mathtt{scan2}} = \sum_{i=1}^n \alpha_i c_i \tag{14}$$

The context vector $h^{\mathtt{scan2}}$ is then fed into a fully-connected layer and a softmax layer to do the classification task.

The whole network is training to minimize the cross-entropy error $E = \frac{1}{N}\sum_{i=1}^N H(y|p) = -\frac{1}{N}\sum_{i=1}^N \sum_{j=1}^n y_j \log p_j$ over all training examples, where $p$ denotes the predictions and $y$ the true labels. Parameters are updated using the RMSProp (Tieleman & Hinton, 2012) rule.

### 3.3 SINGLE-SCAN APPROACH WITH ATTENTION

In the proposed TS-ATT model, we use two Bi-LSTM netwoks to conduct these two scans. However, sometimes we may adopt another faster, but may equally good approach: a single scan. In this way, we first look at the problems to be solved, then scan the original text looking for clues. During this process, we maintain a rough gloal context and pay attention to the important parts simultaneously.

Inspired by this thought, we proposed the **S**ingle-**S**can Approach with **Att**ention (SS-ATT) model, which is a reduced version of TS-ATT model. In this model, the final outputs of the second Bi-LSTM are directly used as attention, saving the need of the first Bi-LSTM network. The SS-ATT model is shown in Figure (2).

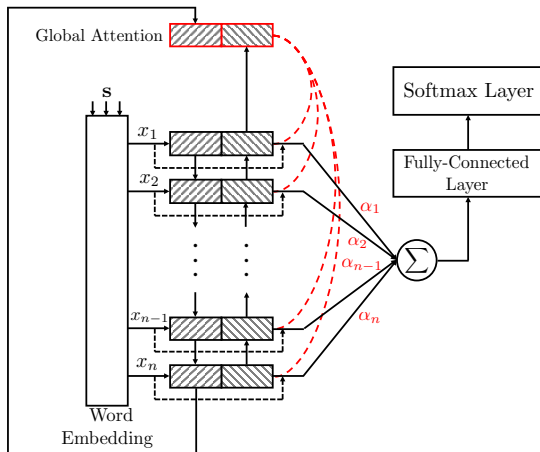

Figure 2: Single-Scan Approach Attention Model (TS-ATT)

## 4 EXPERIMENTS

We evaluated our models on some benchmark datasets for sentiment classification. These datasets includes:

- **Amazon**[1] by Blitzer et al. (2007). Amazon dataset contains product reviews from Amazon.com from many product types. This dataset is origially used for domain adaption, here, we only use it for open-domain classification, regardless of the domain difference.
- **IMDB**[2] by Maas et al. (2011), a large movie review dataset collected from imdb.com website.
- **Yelp 2013**[3] User reviews and recommendations dataset, which are built by Tang et al. (2015).

The statistical information of these datasets are listed in Table (1).

Table 1: Statistical Information of Datasets

| dataset | # classes | # docs | Avg doc length | # words | # Train/(Val/)/Test |
|---|---|---|---|---|---|
| Amazon | 2 | 8,000 | 725 | 42,266 | 6,400 / 1,600 |
| IMDB | 2 | 50,000 | 228 | 88,584 | 25,000 / 25,000 |
| Yelp2013 | 5 | 78,966 | 161 | 73,835 | 62,522 / 7,773 / 8,671 |

[1] https://www.cs.jhu.edu/~mdredze/datasets/sentiment/
[2] http://ai.stanford.edu/~amaas/data/sentiment/
[3] http://ir.hit.edu.cn/~dytang/

For IMDB, the original dataset already has a train/test split, we stick to this behavior in our experiments. For Amazon, we split 20% of the data as the test set. For Yelp datasets, they also have a train/validation/test split.

For datasets without an explicit validation split (like IMDB and Amazon), we randomly select 10% of the training data as the validation data, which is used for hyperparameter tuning and early-stop in training.

## 4.1 BASELINES

We compare our attention model with some models related to our work. These models contain traditional machine learning methods such as support vector machine (SVM) as well as neural models such as LSTM.

**Traditional machine learning methods**:

(1) **SVM+BOW**: Support Vector Machine with Bag Of Words representation. SVM uses hinge loss to maximize the margin between different classes.

(2) **NB+BOW**: Naive Bayes with Bag Of Words representation. NB chooses the class with the largest posterior probability, which is calculated through priori probability and likelihood based on training data.

**Neural model methods**:

(3) **CNN**: Convolutional Neural Network for sentence classification model proposed by Kim (2014). We use 100 filters to evaluation this model.

(4) **LSTM**: Long Short Term Memory network (Graves, 2012)

(5) **Bi-LSTM**: Bi-directional LSTM network (Graves, 2012)

(6) **RCNN**: Recurrent Convolutional Neural Network proposed by Lai et al. (2015), which uses a bi-directional recurrent network to replace the convolution operation in a typical convolutional network.

(7) **NAM**: Neural Attention Model proposed by Shen & Lee (2016). This model uses LSTM output as attention directly for word embeddings.

## 4.2 SETUP

For texts of these datasets, we split them into word tokens using punctuations and spaces as delimiters. Unlike some studies earlier, we process each text as a single unit rather than split them into sentences, which actually provides us more challenges. We process texts this way mainly to test the model's ability to construct representation of long texts, which requires the model to learn long-time dependency.

For the neural models, we set the word embedding dimention to be 300, and LSTM state dimention to be 300. For training, we use stochastic gradient descent based on mini-batches with a batch size of 32. The training process is monitored by the validation accuracy for early-stopping. After training, the model is evaluated by a separate test dataset.

Due to the different pre-processing and model parameters selecting scheme, our results may not look the same as they are reported in other papers. For a fair comparison, we rerun all the models using the same platform of our experiment. No tricks such as dropout, batch-normaliztion are used, since we want a comparison of these models themselves. Similarly, we use random initial word embeddings instead of pretrained embeddings such as word2vec (Mikolov et al., 2013a).

## 4.3 RESULTS AND DISCUSSIONS

Our experimental results are shown in Table (2).

Table 2: Classification Accuracy (in percentage)

| Model | Amazon | IMDB | Yelp2013 |
|---|---|---|---|
| SVM+BOW | 80.31 | 84.88 | - |
| NB+BOW | 80.75 | 82.98 | - |
| CNN (Kim, 2014) | **83.94** | 85.68 | 55.67 |
| LSTM (Graves, 2012) | 79.50 | 82.00 | 56.74 |
| Bi-LSTM (Graves, 2012) | 80.13 | 82.54 | 56.35 |
| RCNN (Lai et al., 2015) | 81.38 | 82.75 | 57.21 |
| NAM (Shen & Lee, 2016) | 79.23 | 86.02 | 57.05 |
| TS-ATT | 82.19 | **86.25** | **58.66** |
| SS-ATT | **83.25** | **86.74** | **58.38** |

As shown in Table (2), our methods (TS-ATT or SS-ATT) achieve the best accuracy on two of three datasets (IMDN and Yelp2013). For Amazon dataset, the result of our method is almost as good as the best model (CNN). These results demonstrate the effectiveness of our proposed models.

From Tabel (2), we can see CNN shows good performance in these sentiment classification tasks. But, CNN has many hyper-parameters to tune, such as filter numbers, filter size, pooling size, etc. Changes of these hyper-parameters affect CNN's performance to a large scale. Zhang & Wallace (2015) did a sensitivity analysis of CNNs for sentence classification. Compared with CNNs, recurrent based networks have fewer hypermeters to fine-tune. Even so, basic neural network such as LSTM doesn't perform so well, as shown in Table (2). Bi-LSTM fails to make significant improvements on these tasks.

Comparing with RCNN, which uses a max-pooling operation after an Bi-LSTM, our model surpass it on all three datasets (1.87%, 3.99% and 1.45% respectively). This result demonstrates the effectiveness of using global context (though a rough one) as attention. Further, our SS-ATT model can be seen as a single Bi-LSTM network with global attention. The great improvement compared to Bi-LSTM also witnesses the key role that global attention plays.

Comparing with NAM, which use global attention directly on word embeddings, our model also achieves big or small improvements (4.02%, 0.72% and 1.61% respectively). These improvements over NAM model indicate that incorporating local contexts with attention would be more suitable for text understanding.

As for our proposed models, we can see TS-ATT and SS-ATT can achieve approximately equally good results. The key difference of these two models is that SS-ATT uses only one Bi-LSTM, improving global and local context representation mutually when training. These results are consistent with our intuition of human-like reading style. Imagine when in a reading comprehension test, we may skip reading the whole texts at first, but directly look at the texts after knowing the questions. In this way, we are actually using single-scan approach to save time. But as long as we have the big picture (global context) in mind, this kind of process won't harm our right decisions.

## 5    ATTENTION VISUALIZATION AND CASE STUDY

In order to get a deeper insight into our global-local context attention model, we track the attention weights when the model is trying to make a decision. Take the IMDB dataset as example, when evaluating our model, we accumulate the attention weights for each word appeared, then calculate their average attentions. By this way, we can get a better knowledge about what kind of words this model favors when trying to predict the sentiment of a text. The words with largest average attention are shown in Table (3). We select those words with a minimal appearing frequency (for IMDB we set it to 100, change of this number won't affect the results so much).

From Table (3), we can see that our attention model can effectively find the most discriminative features. Word such as `surprised` is actually negative in most cases, while in our model it's classified as positive correctly, since we usually use `surprised` to express our pleasance when some movie is beyond our expectation. Also, we notice that the word `recommended` is favored by

Table 3: Words with the largest average attention

| Sentiment | Words with largest average attention in IMDB dataset |
|---|---|
| positive | *recommended, surprised, underrated, excellent, fascinating* |
| | *favorites, incredible, terrific, beautifully, funniest, amazing* |
| negative | *avoid, blah, disappointment, waste, awful, disappointing, recommended* |
| | *dreadful, worst, lousy, miserably, horrible, boredom, pretentious* |

both polarities. In this case, a model has to rely its local context (such as `not recommended` or `recommended`) to make a right decision. This will be illustrated in following case study.

Since our method incorporates local contexts to work, we next provide some case studies where important local contexts are automatically selected for right decisions. We use cases where the word `recommended` serving a high attention and then dive into its surrounding words to support our thought above.

Figure (3) shows the attention weights of each word in the sample sentence segment *or music of the and greeks - this movie is strongly **recommended** for lovers of the music*. Figure (4) shows another negative sentence segment *the new zealand film commission fourthly - a friend **recommended** it to me - however i was utterly disappointed*.

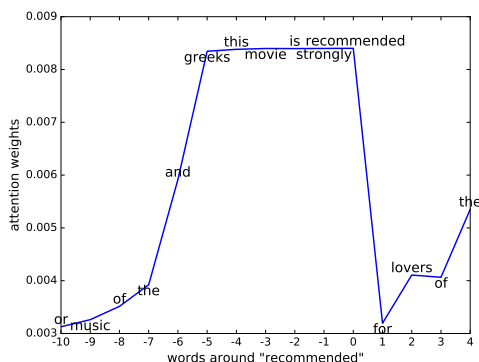

Figure 3: A positive example

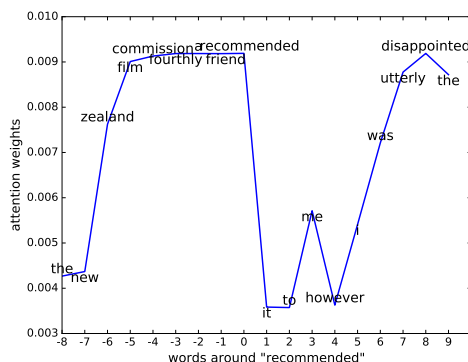

Figure 4: A negtive example

From Figure (3) and Figure (4), we can see that the model actually concentrations on important local contexts rather than single embeddings. This further verifies our initial idea that, using a global context to incorporate local contexts is a suitable way for sentiment or more general text classification task.

## 6 CONCLUSIONS

In this paper, we proposed a global-local context attention framework for sentiment analysis. This method is similar to human's reading behavior in a reading comprehension situation. First, this model uses a Bi-LSTM network to extract a global context representation. This global representation maybe inaccurate, but we only use it as attention for important local parts. In a second scan of the text, the model automatically finds the most important local contexts and incorporates them to make a final decision. A simple version of this model only needs one-scan of text, which can reduce the model complexity but almost equally effective. Experimental results demonstrate that our model performs better than existing related models. Using global context representation as attention, our model can effectively find the most important local parts to make a right decision.

ACKNOWLEDGMENTS

This research is supported by the Key Program of National Natural Science Foundation of China (Grant nos. U1536201 and U1405254), the National Natural Science Foundation of China (Grant no. 61472092), the National High Technology Research and Development Program of China (863 Program) (Grant no. 2015AA020101), the National Science and Technology Support Program of China (Grant no. 2014BAH41B00), and the Initiative Scientific Research Program of Tsinghua University.

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
