# Peer review of "Learning to Understand: Incorporating Local Contexts with Global Attention for Sentiment Classification"

_ICLR 2017 — rejected_

[Official Review · AnonReviewer2 · rating 3 · confidence 4 · 19 Dec 2016]
**official review**

The paper proposes to enhance the attention mechanism for sentiment classification by using global context computed by a Bi-LSTM. The proposed models outperform many existing models in the literature on 3 sentiment analysis datasets. 

The key idea of using Bi-LSTM to compute global context for attention is actually not novel, as proposed several times in the literature, e.g., Luong et al (2015) and Shen & Lee (2016). Especially, Luong et al (2015) already proposed to combine global context with local context for attention.

Regarding to the experiments, of course it would be nice if the model can work well without the need of tricks like dropout or pre-trained word embeddings. However, it would be even better if the model can work well using those tricks. The authors should show results of the models using those tricks and compare them to the results in the literature.  


Ref:
Luong et al. Effective Approaches to Attention-based Neural Machine Translation. EMNLP 2015

[Official Review · AnonReviewer1 · rating 4 · confidence 4 · 19 Dec 2016]
**feedback**

This paper presents a hierarchical attention-based method for document classification. 
The main idea is to first run a bidirectional LSTM to get global context vector, and then run another attention-based bidirectional LSTM that uses the final hidden state from the first pass to weight local context vectors (TS-ATT). 
A simpler architecture that removes the first LSTM and uses the output of the second LSTM as the global context vector is also proposed (SS-ATT). 
Experiments on three datasets are presented, however the results are mostly not state-of-the-art.

I think the idea is nice, but the experiment results are not convincing enough to justify this new model architecture. 
Why is your Yelp 2013 dataset smaller than the original Tang et al, 2015 paper that has ~300k documents? 
I noticed your other datasets are also quite small. Is it because your model is difficult to scale to large datasets?
You should also include results from Tang et al., 2015 in Table 2 that achieves 65.1% accuracy on Yelp 2013 (why is your number so much lower?)
I also suggest removing phrases such as "Learning to Understand" when presenting their model.
Overall, I think that this submission is a better fit for the workshop.

Minor comments:
- gloal -> global
- Not needing a pretrained embeddings, while of course nice, is not that big of a deal. Various models will work just fine without pretrained embeddings.

[Official Review · AnonReviewer3 · rating 3 · confidence 4 · 24 Dec 2016]
**below acceptance threshold**

The authors did not bother responding or fixing any of the pre-review comments. Hence I repeat here:

Please do not make incredibly unscientific statements like this one:
"The working procedure of this model is just like how we human beings read a text and then answer a related question. "
Really, "humans beings" have an LSTM like model to read a text? Can you cite an actual neuroscience paper for such a claim? The answer is no, so please delete such statements from future drafts.

Generally, your experiments are about simple classification and the methods you're competing against are simple models like NB-SVM. So I would change the title, abstract ad introduction accordingly and not attempt hyperbole like "Learning to Understand" in the title.

Lastly, your attention level approach seems similar to dynamic memory networks by Kumar et al. they also have experiments for sentiment and it would be interesting to understand the differences to your model and compare to them.

Other reviewers included further missing related work and fitting this paper into the context of current literature.
Given that no efforts were made to fix the pre-review questions and feedback, I doubt this will become ready in time for publication.

[Final Decision · Program Chairs · 06 Feb 2017]
**ICLR committee final decision**

The consensus amongst reviewers' was that this paper, incorporating global context into classification, is not ready for publication. It provides no novelty over similar methods. The evaluation did not convince most of the reviewers. The paper seems peppered with unjustified and (as rather bluntly, but accurately, put by one reviewer) unscientific claims. Disappointingly, the authors did not respond to pre-review questions. Perhaps more understandably, they did not respond to the uniformly negative reviews of their paper to defend it. I see no reason to diverge from the reviewers' recommendation, and advocate rejection of this paper.